# Implementation and Investigation of an Advanced Induction Machine Field-Oriented Control Strategy Using a New Generation of Inverters Based on dSPACE Hardware

**Mouna Es-saadi** [1,*], **Hamid Chaikhy** [2] **and Mohamed Khafallah** [1]

1   Department of Electrical Engineering, National School of Electricity and Mechanics (ENSEM),
    Hassan II University, Casablanca 20100, Morocco
2   Ecole Nationale des Sciences Appliquees, Universite Chouaid Doukkali, El Jadida 24002, Morocco
*   Correspondence: mounaessaadi2@gmail.com

**Abstract:** Widely used in industrial applications, the induction machine is the subject of many researches. Many are aimed at developing its performances, such torque ripples, current distortions or even rotor speed response, by using different control strategies or even replacing two level inverters in a field oriented control strategy with a new generation of inverters. This paper presents an advanced asynchronous machine field-oriented control strategy with a three level neutral point clamped inverter. The attractive performances of the field oriented control strategy using a three level neutral point clamped inverter are experimentally tested. Both conventional and new field-oriented control strategies are implemented in a dSPACE board induction machine. To highlight the advantages of the new control strategy, conventional and improved strategies are studied in open loop and closed loop conditions using integral proportional and proportional integral controllers, in term of current distortions, torque and speed response.

**Keywords:** dSPACE; new field oriented control strategy; three levels neutral point clamped inverter





## 1. Introduction

Last decade, industries, including the railway traction and automotive industries, have seen a wide improvement. In fact, the railways are one of the best energy-efficient means of mass transportation [1–3]. However, locomotive performances have to be improved in such a way as to meet customer requirements. Accordingly, a new generation of inverters called multilevel inverters is being used to improve railway traction [4–9]. Two level and multilevel inverters have been widely compared in the literature. By dint of the multilevel inverter, efficiency is enhanced in classical driving cycles [10,11]; prices of the two systems are compared in [8,10]. By dint of the reduced filter and battery price for a multilevel drive, the total system is cheaper, even though the cost of a multilevel inverter is slightly higher than a two level inverter. Thanks to multilevel inverters, railway traction can overcome many obstacles, with enhanced performance and less vibration resulting in faster, more pleasant and quieter journeys [12]. The previous researches have investigated topologies. However, another aspect of multilevel traction drives needs to be studied, compared and experimentally tested: performances of control strategies. The target of this work is to present experimentally the enhancement of a standard field oriented control strategy (FOC), and then to show induction machine performances in both steady state and transient context after changing the conventional two level inverter (2 L) by a three level neutral point clamped inverter (3L_NPC) controlled by space pulse width modulation (SPWM). Using a dSPACE 1104 board for a 1.5 kw asynchronous machine, both the 2L_FOC and field oriented control strategy(FOC) using a 3L_NPC (3L_FOC) are experimentally compared, in term of torque ripples, current distortions and rotor speed response for medium and small speeds using integral proportional (IP) and proportional integral controllers (IP). The principles of FOC and 3L_NPC are also introduced.

## 2. Principle of the Conventional and Improved FOC Strategy

### 2.1. Principle of the Conventional FOC Strategy

The aim of the FOC strategy is to have an independent control over the couple and the flux similar to a separately excited DC machine [13–15]. To reach such decoupled control, a FOC algorithm is required to obtain the rotor flux angular position, to correctly align the stator correct vector. Hence, it is feasible to control torque and rotor flux in a DC machine control fashion, by acting on two separated stator current components: $i_{sd}$ and $i_{sq}$. Thanks to the FOC strategy, an asynchronous machine can be used in high-dynamic performance required where only a DC machine can be used

After making the rotor field rotation (Figure 1) $\hat{\psi}_{rd} = \hat{\psi}_r$ and $\hat{\psi}_{rq} = 0$. As presented in Equations (1) and (2) IM relations then becomes:

$$\psi_{rd} = \frac{M}{1 + pT_r} i_{sd}, \tag{1}$$

$$T_e = n_p \frac{M}{L_r} \psi_r i_{sq}, \tag{2}$$

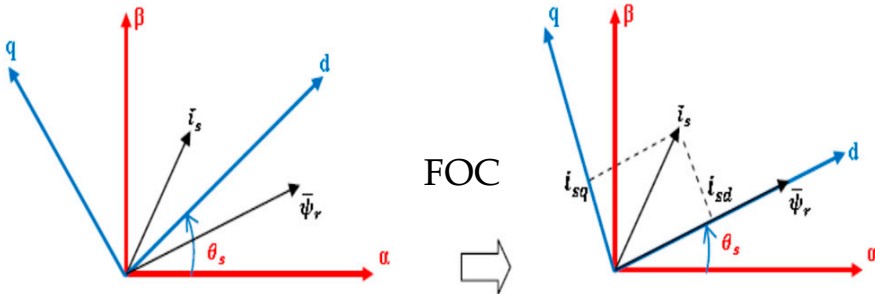

**Figure 1.** Orientation of the rotor field to d axis of the (d,q) reference.

Rotor field rotation is applied once the rotor flux angular position is known. Direct field oriented control (DFOC) and Indirect field oriented control (IFOC) are both employed to obtain this position. Manipulating Equations (3)–(7), $\hat{\psi}_r$ and $\hat{T}_e$ are estimated using $i_{sd}$ and $i_{sq}$, respectively, and are compared to $\psi_r^*$ and $T_e^*$. As shown in Figure 2, torque and flux are directly controlled in DFOC.

$$v_{sd} = \left[ (R_s + p\sigma\mathcal{L}_s) \frac{(1 + T_r p)}{M} + \frac{M}{\mathcal{L}_r p} \right] \psi_{rd} - w_s \mathcal{L}_s \sigma i_{sq}, \tag{3}$$

$$v_{sq} = (R_s + p\sigma\mathcal{L}_s) \frac{T_e}{p\frac{M}{L_r}\psi_{rd}} + w_s \mathcal{L}_s \sigma i_{sd} + w_s \frac{M}{\mathcal{L}_r} \psi_r, \tag{4}$$

$$\hat{\psi}_r = \frac{M}{1 + pT_r} i_{sd}, \tag{5}$$

$$\theta_s = \int \left( n_p \Omega + \frac{M}{T_r} \frac{i_{sq}^*}{\psi_{rd}^*} \right) dt \, \hat{T}_e = n_p \frac{M}{L_r} \psi_r i_{sq}, \tag{6}$$

Figure 2 presents the control strategy scheme of DFOC. This strategy is chosen for this work and will be named the FOC strategy.

To secure decoupled control for torque and flux, compensation terms $f_{emd}$ and $f_{emq}$ Equations (7) and (8) are added to obtain d and q axes completely independently.

$$f_{emd} = w_s \mathcal{L}_s \sigma i_{sq} \tag{7}$$

$$f_{emq} = w_s \mathcal{L}_s \sigma i_{sd} + w_s \frac{M}{\mathcal{L}_r} \psi_r \tag{8}$$

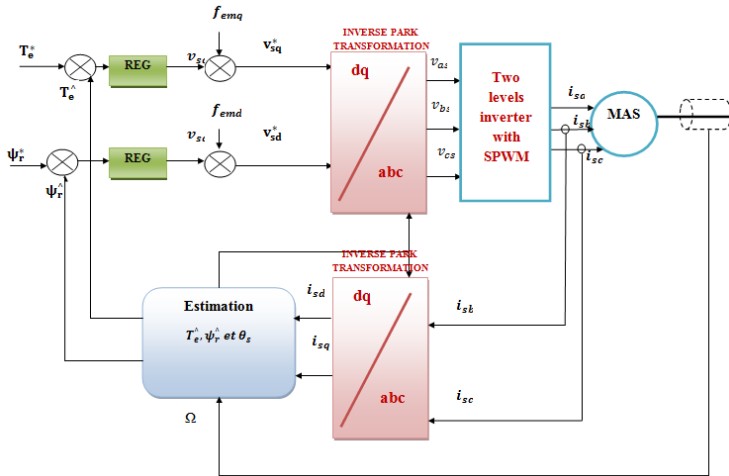

**Figure 2.** DFOC strategy scheme of asynchronous machine using 2 L.

Equation (2) presents how position $\theta_s$ and voltages $v_{sq}$, $v_{sd}$ arecalculated. Using park position $\theta_s$, $v_{sq}$, and $v_{sd}$ are changed and injected to the 2 L. The latter is controlled by SPWM. To obtain two voltage levels, SPWM is obtained by comparing a sinusoidal reference signal and triangular carrier. Thereby, the inverter's switches are controlled through pulses obtained by this comparison, as shown in Figure 3. The following Figure 4 shows an experimental result of the two level output voltage inverter implemented using dSPACE 1104 where we observed the two voltage levels.

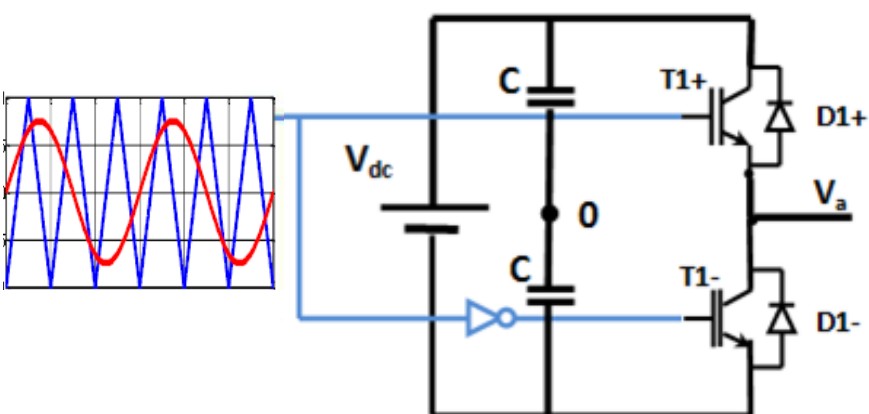

**Figure 3.** Principle of the SPWM two level control strategy.

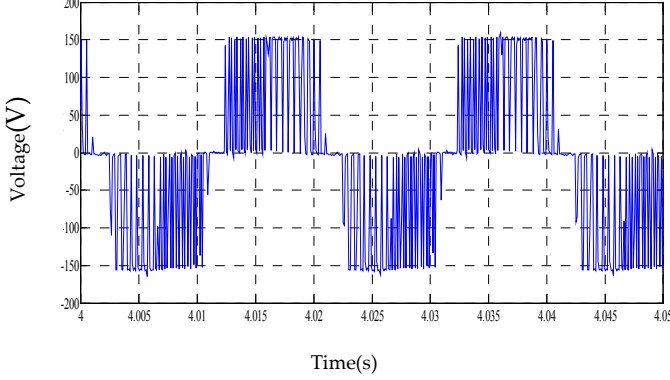

**Figure 4.** Output voltages of the two level inverter.

### 2.2. Principle of the Improved FOC Strategy

Thanks to multilevel conversion structures, medium voltage high-power drives as railway traction drives are improved; the price of semiconductor devices is augmented. Multilevel topologies decrease voltage stress, which compensates for the rising number of devices [16,17]. Additionally, these structures reduce the total harmonic content, and then present the benefit of lowering the volume of the output filter [18]. Thereby, torque ripples will be decreased for motor drive applications. The NPC inverter is the multilevel inverter selected to replace the 2 L [19,20].

#### 2.2.1. Principle of 3L_NPC Inverter

As presented in Figure 5, six clamped diodes and four vertical IGBTs form each leg of the 3L_NPC inverter. As given in Table 1, switching states gives switches that generate five voltage levels. Figure 6 presents the SPWM control strategy using a NPC inverter [21,22].

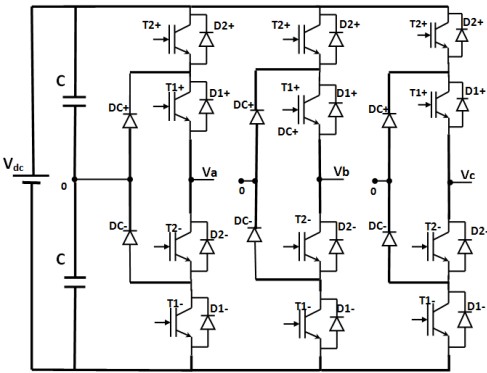

**Figure 5.** Structure of 3L_ NPC.

**Table 1.** Basic switching of 3L_ NPC.

| Voltage Va | Switching State | | | |
|---|---|---|---|---|
| | T2+ | T1+ | T2− | T1− |
| Vdc/2 | 1 | 1 | 0 | 0 |
| 0 | 0 | 1 | 1 | 0 |
| −Vdc/2 | 0 | 0 | 1 | 1 |

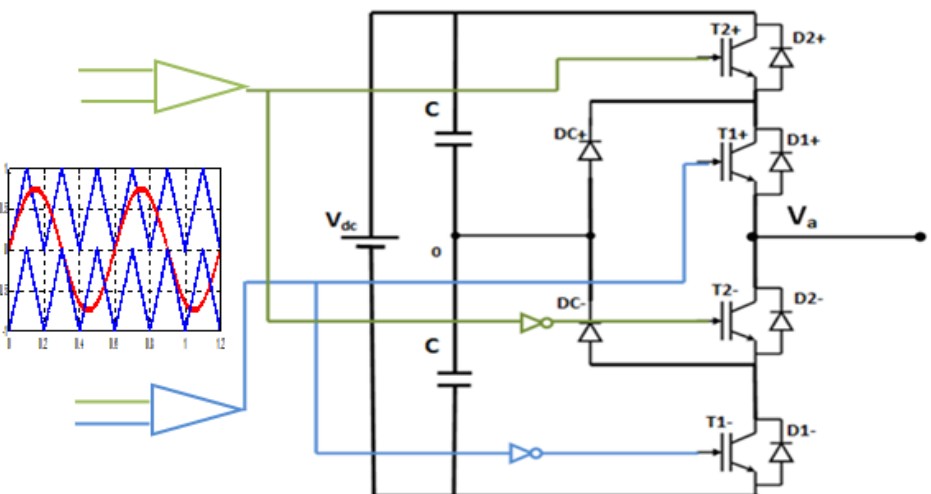

**Figure 6.** Structure of the SPWM for 3L_NPC inverter.

The following Figure 7 shows an experimental result of the output voltages of a 3L_NPC inverter implemented using dSPACE 1104 where three voltage levels are observed.

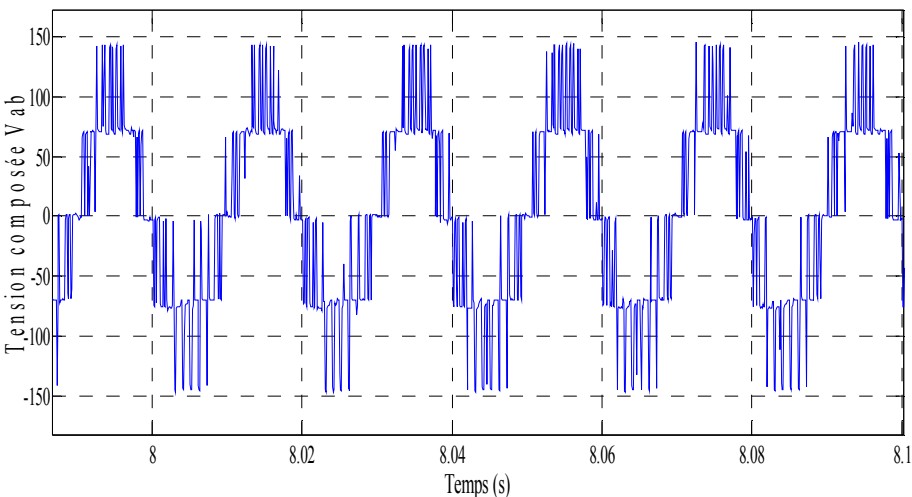

**Figure 7.** 3L_NPC inverter output voltages.

### 2.2.2. Amelioration of FOC Strategy Using 3L_NPC Inverter

To enhance the transient and steady state performances of the FOC strategy for an asynchronous machine, and to decrease voltage stress in semi conductors, particularly for high power applications, with a suitable SPWM control strategy, a 3L_NPC inverter can replace a 2 L inverter. As Figure 8 shows, all the other blocks of the FOC strategy scheme are still the same [23].

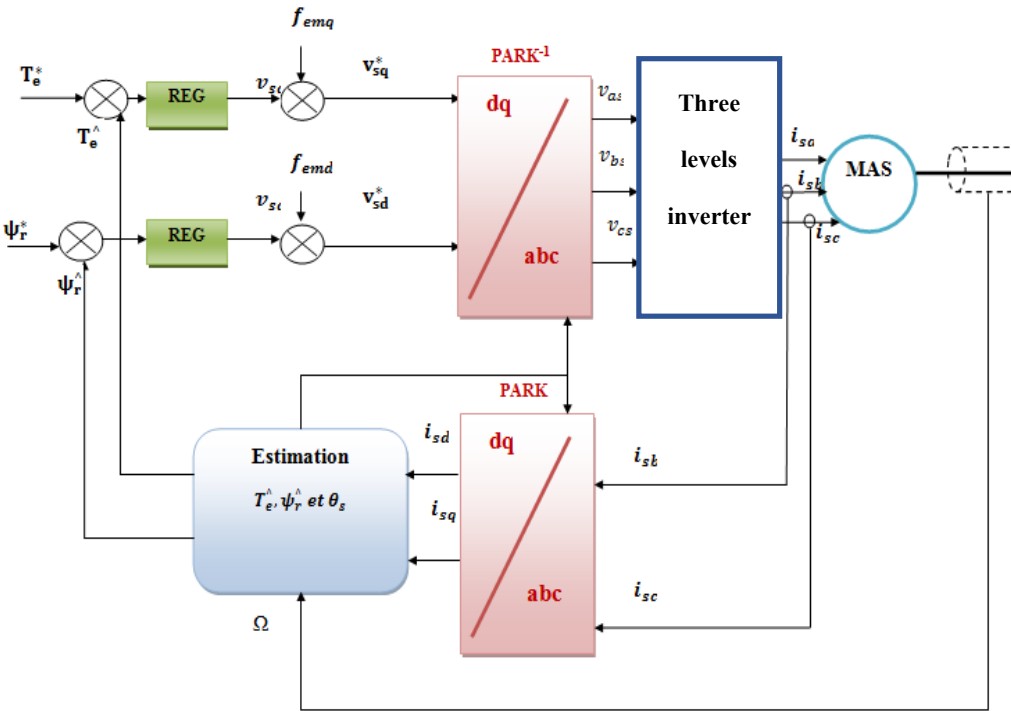

**Figure 8.** FOC of IM using 3L_NPC inverter.

### 2.2.3. Closed Loop Conditions

The rotor speed closed loop DFOC system is shown in Figure 9. The PI and IP controller scheme are, respectively, given in Figures 10 and 11.

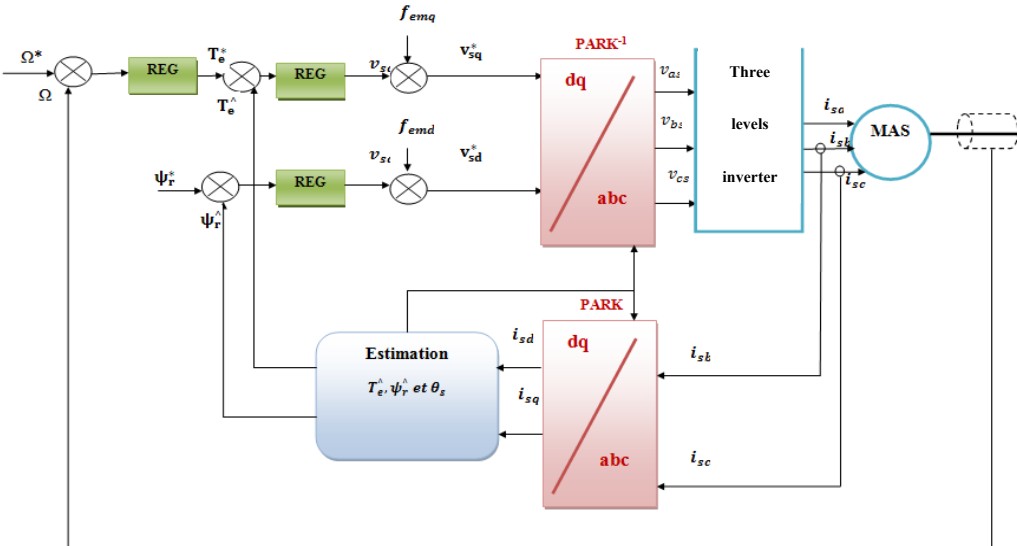

**Figure 9.** Rotor speed closed loop 3L_FOC IM system.

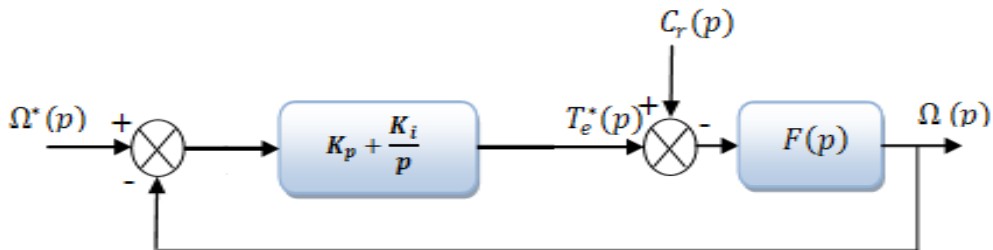

**Figure 10.** Closed loop PI speed controller.

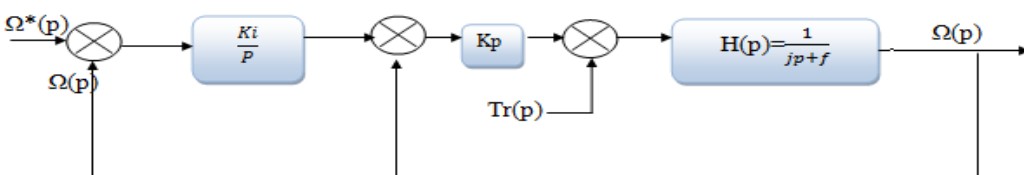

**Figure 11.** Closed loop IP speed controller.

To show an aperiodic representation of the previous presented system, we find following Equations (9) and (10):

$$K_{iip} = \frac{1}{4g\tau}, \quad K_{pip} = \frac{J - \tau f}{\tau}, \tag{9}$$

$$\tau = \frac{J}{K_{pip} + f}, \quad g = \frac{K_p}{K_{pip} + f}, \tag{10}$$

In order to have an aperiodic representation of the above presented system, we find for the PI speed controller the following (Equation (11)).

$$K_{ipi} = J\omega_0^2, \quad K_{ppi} = 2J\omega_0 - f\,J\omega_0^2, \tag{11}$$

Values of all those coefficients are given in Table 2.

**Table 2.** Experimental and machine parameters.

|  | Parameters | Signification |
|---|---|---|
| IM parameters | Rs = 5.63 | Stator Resistance ($\Omega$) |
|  | Rr = 2.62 | Rotor Resistance ($\Omega$) |
|  | M = 0.364 | Mutual inductance (H) |
|  | Ls = 0.382 | Stator inductance (H) |
|  | Lr = 0.382 | Rotor inductance (H) |
|  | J = 0.023 | Factor of inertia(Kg·m$^2$) |
| Experimental parameters | f = 0.00155 | Coefficient of friction |
|  | $p$ = 2 | Number of pole pairs |
|  | $p$ = 1.5 | Rated power (KW) |
|  | Vdc = 150 | DC bus voltage (V) |
|  | Fpwm = 2000 | Pulse width modulation frequency (Hz) |
|  | F = 10 | Sampling frequency (KHz) |
|  | Kppi = 0.59, Kipi = 2.3 | Integral and proportional coefficients for PI controller |
|  | Kpip = 0.297, Kiip = 6.01 | Integral and proportional coefficients for IP controller |

## 3. Experimental Results

*Presentation of the Experimental Platform*

Structure and photography of the experimental platform are presented in Figures 12 and 13, respectively, and contain: a 3L_NPC inverter, the 1.5 KW asynchronous machine (coupled in star) is driven under load using the help of a DC generator mechanically coupled to the motor with the following parameters: 1 KW, 220 V, 6.5 A, 2520 rpm. The latter furnishes a 4KW resistive bank to give a number of load torques, a dSPACE 1104 board founded on a 250 MHz 603-PowerPC- 64-bit processor and a slave-DSP based on a 20 MHz TMS320F240-16-bit microcontroller are employed. The dSPACE operates on a MATLAB/SimulinkR2013b platform. The dSPACE board is working with Control Desk software which realizes the record of the results more easily [24–27], ameliorates the controller and automates the experiments. Using dSPACE 1104, the drive can be designed in MATLAB/SimulinkR2013b and converted to real-time codes using Real-Time Workshop (RTW); for the speed sensor (15 V for 1500 rpm) a tachymeter is used [28–32].

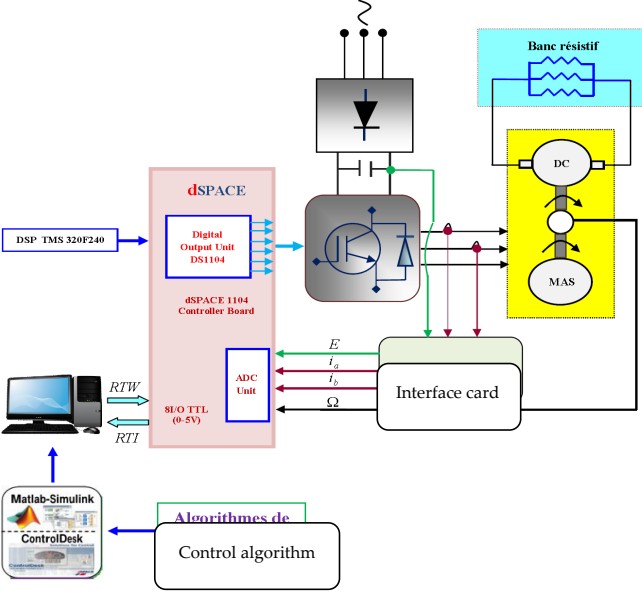

**Figure 12.** Structure of the experimental platform.

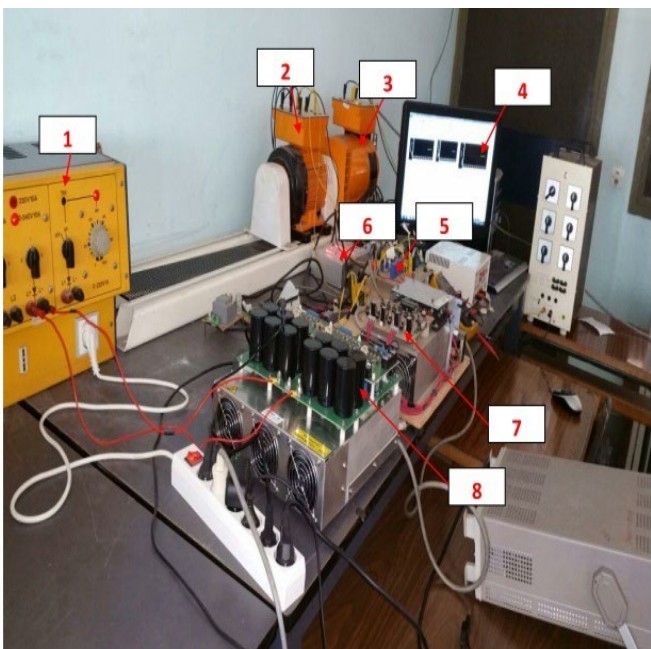

1: DC bus voltage

2: DC machine

3: IM

4: Control desk

5: Interface card

6: dSPACE1104
Input/output interface

7: Two level inverter

8: Three level NPC
inverter

**Figure 13.** Photography of the experimental platform.

To show the performances of the 3L_FOC, different experiments were made with parameters presented in Table 2.

## 4. Experimental Results and Discussion

### 4.1. Open Loop Conditions

As presented in Figure 14, at open loop conditions both 2L_FOC and 3L_FOC torques follow perfectly their references; however, the 2L_FOC presents more torque ripples, traduced by important audible noise and important current distortions (Figure 15), which is totally eliminated in the 3L_FOC.Moreover, as shown in Figure 16, the 3L_FOC has better current THD than the 2L_FOC.

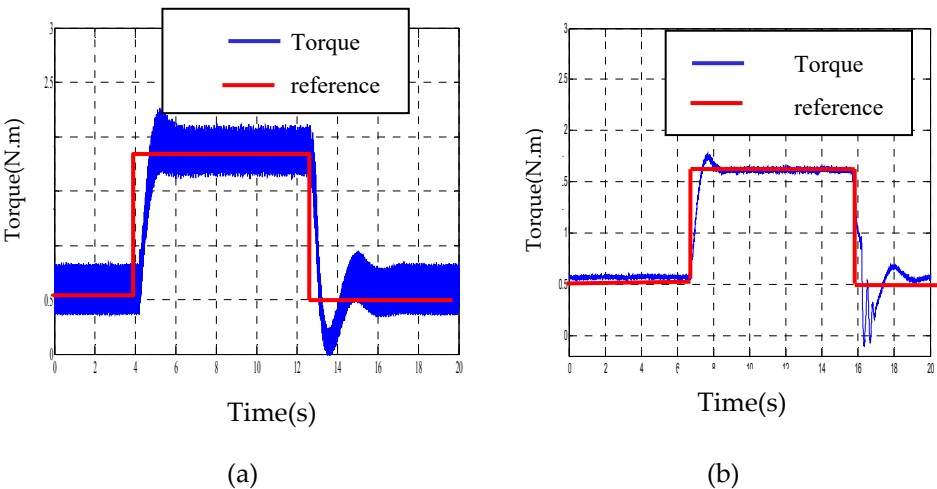

(a)                                        (b)

**Figure 14.** Comparison between (**a**) 2L_FOC and (**b**) 3L_FOC in open and closed loop conditions in terms of Torque ripple.

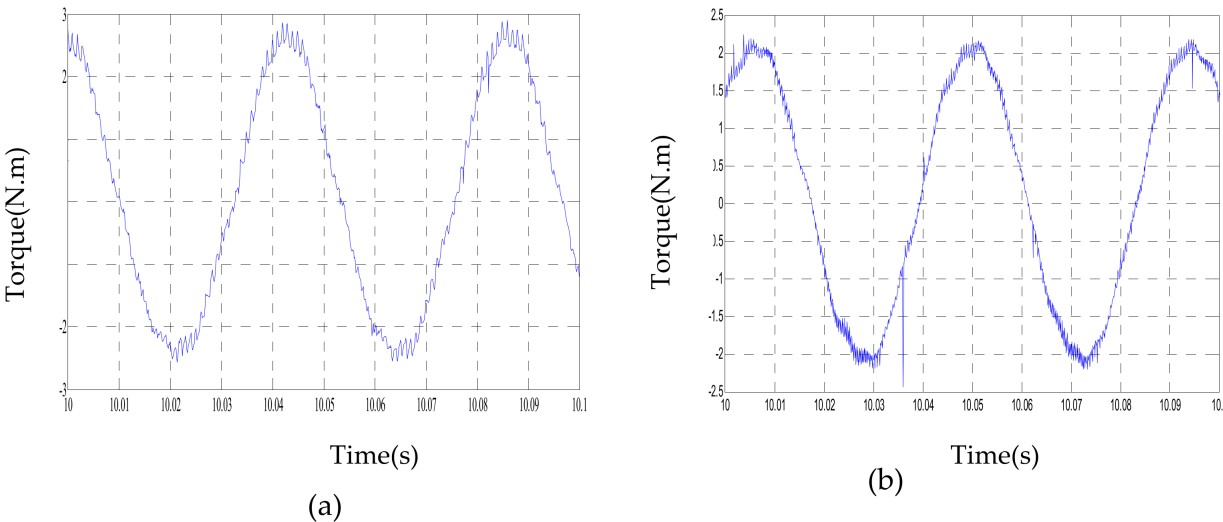

**Figure 15.** Comparison between (**a**) 2L_FOC and (**b**) 3L_FOC in open and closed loop conditions in terms of Current distortions.

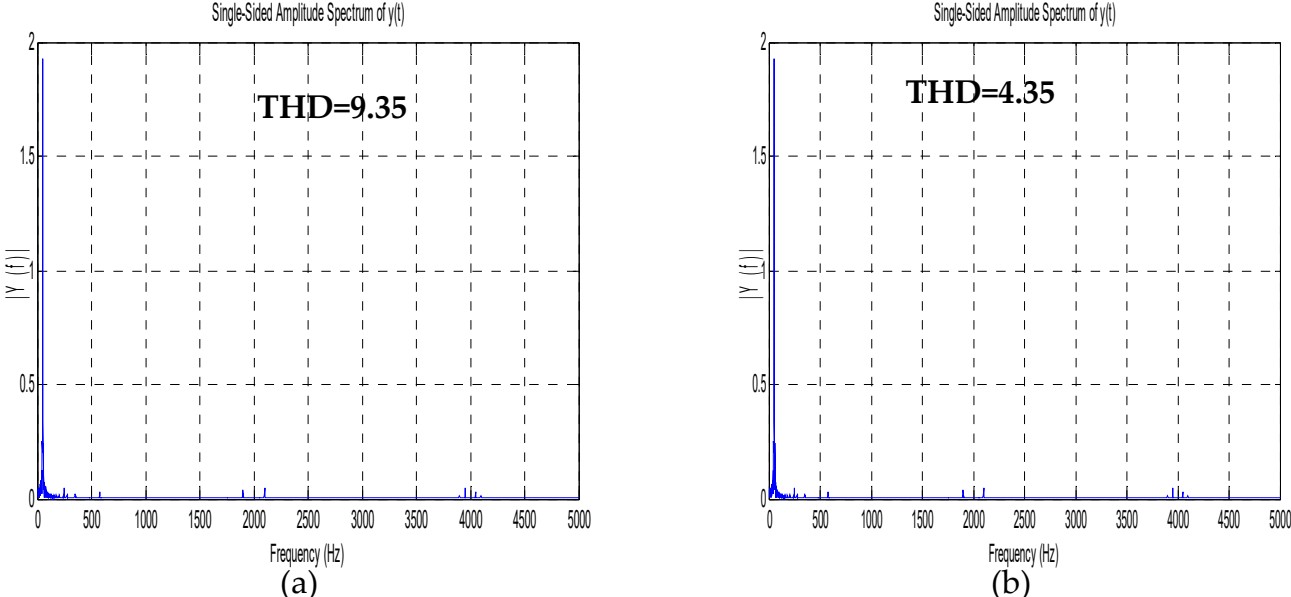

**Figure 16.** Comparison between 2L_FOC and 3L_FOC in open and closed loop conditions in terms of: (**a**) Current THD for 2L_FOC; (**b**) Current THD for 2L_FOC.

### 4.2. Closed Loop Conditions

As presented in Figure 16, at closed loop conditions using a PI and an IP controller, practical tests are made for the 3L_FOC for medium speed with changing direction as shown in Figure 17a,b, with injecting a perturbation as presented in Figure 17c,d, for small speed with changing direction as shown in Figure 17e,f, with injecting a perturbation as presented in Figure 18a,b.Those tests shows that the speed of the 3L_FOC has excellent performances in term of pursuit and rejection of perturbation. However, the 3L_FOC using an IP controller is still better in term of speed pursuit than a PI controller the latter also presents a peak, absent in the case of the IP controller.

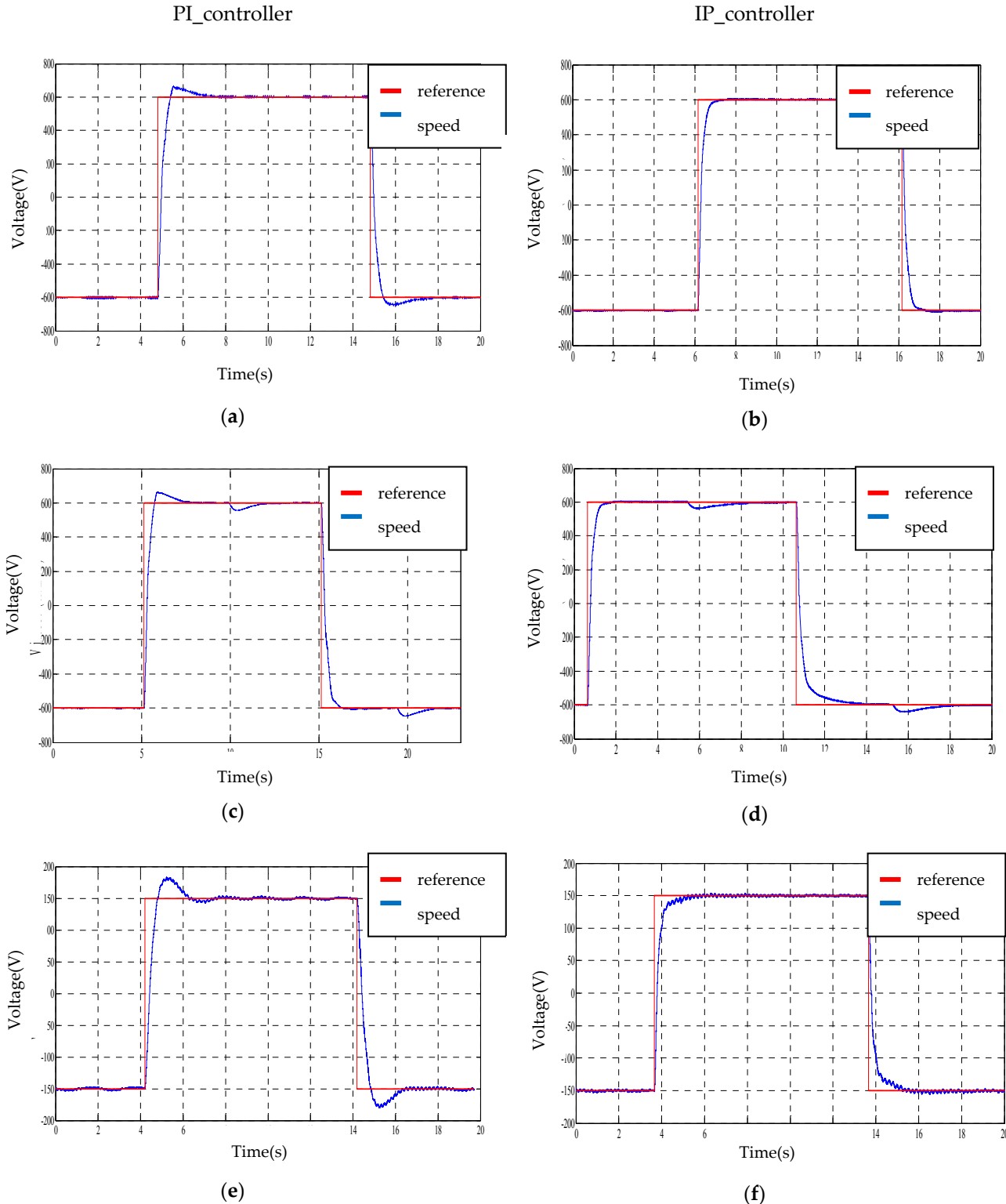

**Figure 17.** Comparison between 3L_FOC in closed loop conditions using PI and IP controller in terms of: (**a**,**b**) Medium speed response with changing direction; (**c**,**d**) Medium speed response with injecting perturbation; (**e**,**f**) Small speed response with changing direction.

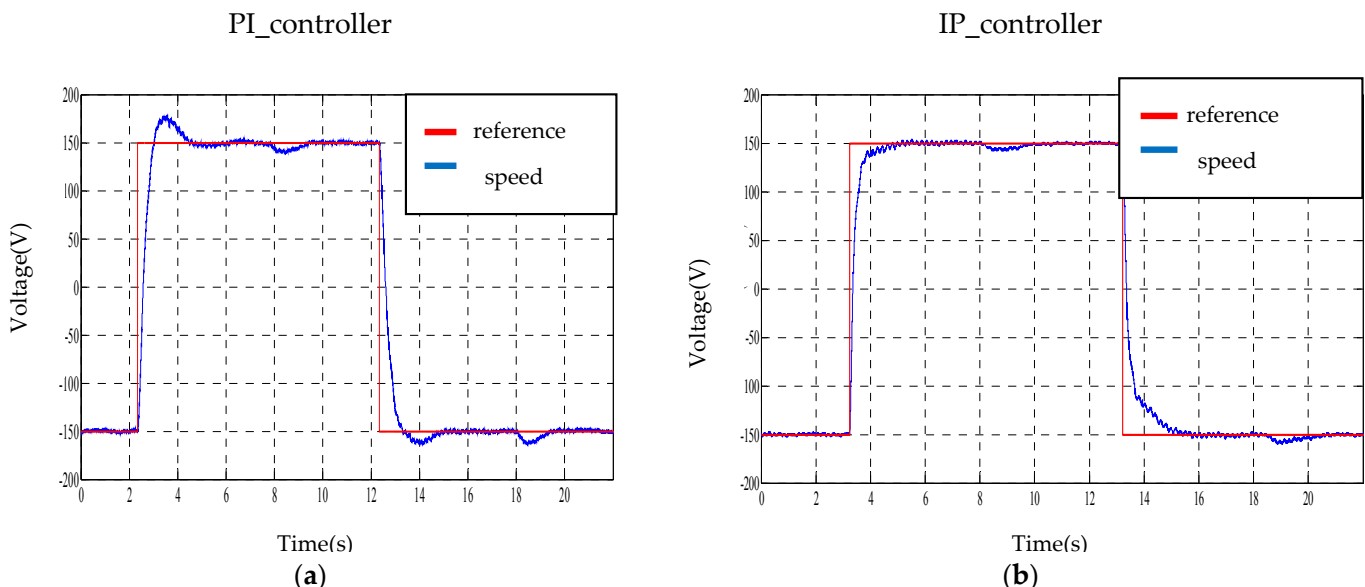

**Figure 18.** Comparison between 3L_FOC in closed loop conditions using PI and IP controller in term of: (**a**,**b**) Small speed response with injection of perturbation.

## 5. Conclusions

Implementation and detailed investigation of an advanced FOC of IM have been presented in this work, using a 3L_NPC. DSPACE hardware has been used to obtain experimental results. Those results validate interesting performances of 3L_FOC in open loop conditions, in terms of current distortions and torque ripple which is clearly traduced by the absence of noise in a 3L_FOC. In closed loop conditions, a detailed investigation is also presented and shows excellent performances of the 3L_FOC in terms of speed pursuit and rejection of perturbation for both a PI and an IP controller, even if the IP controller is still better in terms of speed pursuit. Thus, a 3L_FOC can widely improve overall a railway traction system and make passengers' journeys better. Thereby, in the near future the developed system could be integrated into a new generation of locomotives.

To provide perspective for this work, a higher level multilevel inverter could be used for better performance, eventually controlled by a space vector modulation (SVM).

**Author Contributions:** Methodology, M.E.-s.; software, M.E.-s.; writing, M.E.-s.; investigation, M.E.-s., H.C.; validation, M.E.-s., H.C., funding acquisition, M.E.-s., H.C., M.K.; supervision, M.K.; project administration, M.K. All authors have read and agreed to the published version of the manuscript.

**Funding:** This research received no external funding.

**Conflicts of Interest:** The authors declare no conflict of interest.

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
