# Peer review of "Implementation and Investigation of an Advanced Induction Machine Field-Oriented Control Strategy Using a New Generation of Inverters Based on dSPACE Hardware"

_asi, doi:10.3390/asi5060106_

Round 1
Reviewer 1 Report
I read through the manuscript and did not find any obvious problems. As a scientific paper, I have not found any good innovations. In addition, I did not see the author's organization and address. The authenticity of the paper cannot be determined through network retrieval. That is why I think the manuscript should be major revised.
Author Response
Dear reviewer,
Thank you for your suggestions, we have modified our paper as your comments, and marked in this paper.

Reviewer 2 Report
The title of the paper announces an important objective - Implementation and investigation of an advanced induction machine Field Oriented Control strategy using a new generation of inverters based on dSPACE hardware. Unfortunately, the work turns into a simple "proof of concept" exercise for using the Field oriented control strategy using a three levels Neutral point clamped inverter, with facility given by dSpace.
From my opinion in order to reshape the structure of article it is necessary to include a section – Related work where to describe more specific the other approaches about your subject and try to define some differentials.
In actual section 2.1 it is necessary to review entire flow of equations because few numbers are missing (like 8,9) or appears some inconsistency (eq. 2 from line 89).
The Figure 3 is not cited in the text.
The figure 12 it is not very clear…maybe is an reinterpretation of a figure from a paper, which is not cited in the reference list:
Boulghasoul, Z., Kandoussi, Z., Elbacha, A. et al. Fuzzy Improvement on Luenberger Observer Based Induction Motor Parameters Estimation for High Performances Sensorless Drive. J. Electr. Eng. Technol. 15, 2179–2197 (2020). https://doi.org/10.1007/s42835-020-00495-6
In the figures with experimental results with red line is the reference and with blue is the speed.
One of the main conclusions from section 4.2 about the better performance of IP controller comparatively with PI controller it is classical issue about peak overshoot, not in link with 3L_FOC.
Author Response

(The authors gave the same response as above.)

Reviewer 3 Report
The article presents a new method of controlling an induction machine, using a new generation of inverters. The effect of the presented works is to increase the efficiency of the induction machines. The research methodology described in the paper contains the required information as well as the research area. The presented topic is up to date and may have the practical use. The work is well organized with the introduction, the theoretical part,the presentation of the results of the interviews and the conclusions. After the theoretical part, the paper presents an analysis of the research results. Each part is presented correctly and is correlated with the rest of the article.
Recommendations for improving the manuscript:
1. There is a noticeable lack of a detailed description of DSpace 1104. Who is its developer?
2. There is no reference to Figure 3.
3. Does chapter 2.2.2. have to be as a separate chapter? It is very short.
4. A more detailed description of Figures 14-17 is suggested.
5. What are the plans for further research?
6. What is the practical application of the presented results?
Author Response

(The authors gave the same response as above.)

Round 2
Reviewer 2 Report
Were made a few changes asked.
Figure 12 must be renewed.
Author Response
Dear reviewer.
Thank you for your professional suggestions.
We wish to submit a new manuscript entitled “implementation and investigation of an advanced induction machine field oriented control strategy using a new generation of inverters based on dSPACE hardware” for consideration by the Applied system innovation. We confirm that this work is original and has not been published elsewhere nor is it currently under consideration for publication elsewhere.
In this final version of paper, we revise the manuscript according to the referees’ comments, we replace FIG.1 by figure.1, we check all referees’ comments and modifications and we approved them.
Please address all correspondence concerning this manuscript to me.
Thank you for your consideration of this manuscript.
Sincerely,
Mouna ES-SAADI
